# A Capacitive MEMS Inclinometer Sensor with Wide Dynamic Range and Improved Sensitivity

**DOI:** 10.3390/s20133711

**Published:** 2020-07-02

**Authors:** HanYang Xu, Yulong Zhao, Kai Zhang, Kyle Jiang

**Affiliations:** 1State Key Laboratory for Manufacturing Systems Engineering, Xi’an Jiaotong University, Xi’an 710049, China; xuhanyang@stu.xjtu.edu.cn (H.X.); zhangkai1995@stu.xjtu.edu.cn (K.Z.); 2School of Mechanical Engineering, University of Birmingham, Birmingham B15 2TT, UK; k.jiang@bham.ac.uk

**Keywords:** liquid MEMS inclinometer sensor, array capacitor, super-hydrophobic, wide measurement range

## Abstract

This paper proposes a novel capacitive liquid metal microelectromechanical system (MEMS) inclinometer sensor and introduces its design, fabrication, and signal measurement. The sensor was constructed using three-layer substrates. A conductive liquid droplet was rolled along an annular groove of the intermediate substrate to reflect angular displacement, and capacitors were used to detect the position of the droplet. The numerical simulation work provides the working principle and structural design of the sensor, and the fabrication process of the sensor was proposed. Furthermore, the static capacitance test and the dynamic signal test were designed. The sensor had a wide measurement range from ±2.12° to ±360°, and the resolution of the sensor was 0.4°. This sensor further expands the measurement range of the previous liquid droplet MEMS inclinometer sensors.

## 1. Introduction

Metal liquid droplets with high surface tension and electrical conductivity have attracted significant interest in various advanced microelectromechanical system (MEMS) sensor applications such as the droplet acceleration sensors [1], liquid touch sensors [2,3,4], and liquid droplet inclinometer sensors [5,6]. Compared with solid-state beam sensors, MEMS sensors that use metal droplets as sensitive elements have the advantage of having a high resistance ability for overload [7,8,9]. Droplets will only be squeezed and deformed under a large acceleration and will soon return to the initial shape after the acceleration is released [10]. By taking advantage of this, Park first proposed a MEMS liquid droplet inclinometer sensor with a mercury droplet as a sensitive element [11]. The diamond-shaped groove on the glass substrate provides a channel for the droplet, and the angle signals were estimated when the mercury droplet is connected to two parallel electrodes. However, due to the difficult sliding of the droplet caused by the squeeze deformation of the droplet in the diamond-shaped channel, the sensor was only used in the special angle measurement (12° and 30°). Then, Xu proposed an improved liquid droplet inclinometer sensor, which increased the height of the channel. The droplet is only connected to the lower surface of the groove and can slide well under the action of gravity [12]. The range of measurement angle of the sensor was ±45° and the resolution was 3.625°. However, the acquisition method of the angle signal is to conduct the array electrodes with the droplet, which makes it necessary to manufacture a large number of metal electrodes to improve the resolution of the sensor. Meanwhile, increasing the sensitivity of the sensor requires reducing the sliding angle of the metal droplet, which causes the contact area between the droplet and the metal electrodes to be reduced and the conduction fails. Hence, an inclination sensor capable of outputting angle signals without causing the metal droplet to connect with the electrodes is necessary.

MEMS capacitance sensors mainly consist of capacitor electrodes and insulation medium. The amount of capacitance is related to the distance and the area between two electrodes [13,14]. In the MEMS fabrication process, the capacitors can be easily fabricated by using sputtering or plating methods [15]. Hence, these are widely adopted for measuring acceleration, displacement, and angular signals [16,17,18]. Since the capacitive sensor does not require the sensitive element to be connected to the electrodes, the issue of conduction failure of the droplet inclinometer sensor is effectively avoided. Furthermore, using array capacitors and appropriately increasing the number of capacitor electrodes can increase the measurement range. Such sensors can be used for signal testing in many complex situations [3,19]. In 2015, Won used a capacitor array to measure the touch signal and the sensor had a large dynamic range (~100 pF), higher sensitivity (~147 pF/N), and relatively good spatial resolution [2].

This paper proposed a MEMS liquid droplet inclinometer that measures angle signals with an array of capacitors. The use of capacitor arrays to obtain angular displacement in the MEMS droplet inclinometer sensor can receive continuous signals that the previous droplet sensors cannot. The COMSOL numerical simulation software was used to analyze the change of capacitance of each capacitor electrode after the droplet passed through the capacitor electrodes, and the overall structure of the sensor was designed according to the results of the numerical simulation. After completing the manufacturing process, the dynamic and static output signals of the sensor were tested. The sensor had a large measurement range of ±2.12° to ±360°, and the resolution of the sensor was 0.4°. This research has greatly improved the measuring range and the resolution of MEMS liquid inclinometer sensors.

## 2. Device Concept and Design

### 2.1. Structure of the Sensor

The schematic diagram of the MEMS liquid droplet sensor structure is shown in Figure 1. The sensor consists of a three-layer substrate that includes an upper layer, an intermediate substrate, and a lower substrate. Eighteen array capacitor electrodes and a ring capacitor electrode were deposited on the surface of the lower substrate. Array micro-cubes, which provide the super hydrophobic surface of the liquid droplet, were situated in between the ring capacitor electrode and array capacitor electrodes. Furthermore, an annular groove with a width of 2 mm and a height of 2 mm in the intermediate substrate controlled the movement of the droplet and provided a sliding channel for the mercury droplet. Since the array micro-cubes on the lower substrate provide a super hydrophobic surface, the contact angle between the droplet and the lower substrate surface is increased, and the contact area is decreased [20]. Therefore, no short circuits or signal errors will occur on the capacitor electrodes during the sliding process of the droplet. To avoid the connection between the droplet and the capacitor electrodes caused by the droplet deformation under the action of the large acceleration, a layer of silicon dioxide as an insulating layer on the ring capacitor electrode and array capacitor electrodes of the lower substrate was deposited. In addition, the upper layer encapsulate the entire sensor and left a gap with the liquid droplet, so that the liquid droplet was not squeezed in the annular groove and could slide easily.

### 2.2. Working Principle

The MEMS liquid inclinometer sensor was proposed based on the principle of changing the mutual capacitance between the ring capacitor and the array capacitors by the sliding of the liquid droplet. The working principle of the sensor is as follows:

Figure 2 shows the distribution of the array capacitor electrodes and the ring capacitor electrode. There were 18 array capacitors was 18 that were evenly distributed inside the large ring of the annular groove, and the angle between two adjacent capacitor electrodes was 20°. Moreover, the ring capacitor was deposited outside the small ring of the annular groove. During the measurement process, the sensor was placed in the initial measurement position, and the angle between the sensor and the horizontal plane was *β*. The input angle was provided by rotating the sensor around the horizontal axis and the droplet slid under the action of gravity due to the change in the input angle. The changes in the mutual capacitance between the array capacitor electrodes and the ring capacitor electrode were measured by the new position of the droplet in the annular groove after sliding. Since the sensor was not placed on the horizontal plane, the output angle was not numerically equal to the input angle. The relationship between the input angle and the output angle was analyzed based on the projection of the droplet motion trajectory on the vertical plane [12].

The movement trajectory of the droplet in the sensor is mainly controlled by the annular groove on the intermediate substrate. Here, the shape of the annular groove was designed as a standard circle with a radius of one to facilitate calculation. In addition, the projection of the circular curve on the vertical plane was an ellipse, which satisfies the standard ellipse equation, as shown in Figure 3a.

The standard circle equation of the annular groove on the intermediate substrate of the sensor in coordinate system 1 (X−Y) is shown in Equation (1):(1)$$x^{2}+y^{2}=1$$

Here, *x* and *y* are the horizontal coordinate and vertical coordinate of the point on the circle equation in coordinate system 1, respectively. Furthermore, the elliptic curve in the coordinate system 2 (X_1_–Y_1_) is shown in Equation (2):(2)$$\frac{x_{1}{}^{2}}{a_{1}^{2}}+\frac{y_{1}{}^{2}}{b_{1}^{2}}=1$$

Here, *a*_1_ and *b*_1_ are the long axis and short axis of the elliptic curve projected in the coordinate system 2, *a*_1_ = 1, *b*_1_ = sin*β*, *x*_1_, *y*_1_ are the horizontal and vertical coordinates of the point on the elliptic equation of the coordinate system 2. *β* is the angle between the horizontal plane and the initial position of the sensor.

With the combined action of gravity and surface tension, the metal droplet stayed at the lowest point A of the circular curve (i.e., the lowest point of annular groove on the intermediate substrate) in the initial state. At the same time, the corresponding position of the droplet on the vertically projected elliptic curve was A_1_, as shown in Figure 3a. With input angle *α*, the metal droplet will eventually stop at the lowest point B of the (X_B_, Y_B_) coordinate system 1. The projection point of point B on the elliptic curve was B_1_ (X_B1_, Y_B1_), as shown in Figure 3b. *L_b_*_1_ is the tangent of point B_1_ on the elliptic curve, and the slope of the tangent *L_b_*_1_ in the coordinate system is *tanα*, as shown in Equation (3).
(3)$$\tan\alpha =\frac{x_{B_{1}}b_{1}^{2}}{y_{B_{1}}a_{1}^{2}}$$

When the input angle is *α*, the angle at which the droplet slides in the annular groove of the intermediate substrate is *α*_0,_ and the relationship between *tanα* and *tanα*_0_ is shown in Equation (4):(4)$$\tan\alpha_{0}=\sin\beta \tan\alpha$$
where *α*_0_ is the output angle; *β* is the angle between the sensor and horizontal plane; and *α* is the input angle.

Figure 4 shows the schematic diagram of the droplet leaving and approaching from the capacitor electrodes. The measured output angle signal was represented by measuring the mutual capacitance between each array capacitor electrode and the ring capacitor electrode. When the mercury liquid droplet was far from the capacitor electrodes, as shown in Figure 4a, the amount of capacitance tested by the system was the capacitance formed by the array capacitor electrode and the ring capacitor electrode (named *C*_1_). When the droplet is close to the capacitor electrode, as shown in Figure 4b, the capacitance between the two electrodes will change from *C*_1_ to *C*_2_.

The droplet passing between the two capacitor electrodes is equivalent to reducing the distance between the ring capacitor and the array capacitor, therefore, the mutual capacitance of the ring capacitor and the array capacitors increases as the mercury droplet approaches the capacitor electrodes, and decreases as the mercury droplet moves away from the capacitor electrodes. The mutual capacitance matrix of the sensor is shown in Equation (5):(5)$$C_{mutual}=\left[\begin{array}{cccc} C_{1}^{1} & C_{2}^{1} & \cdots & C_{19}^{1}\\ C_{1}^{2} & C_{2}^{2} & \cdots & C_{19}^{1}\\ \vdots & \vdots & \ddots & \vdots \\ C_{1}^{19} & C_{2}^{19} & \cdots & C_{19}^{19} \end{array}\right]$$

Here, *C_mutual_* is the mutual capacitance matrix of the sensor; *C*_1_~*C*_18_ represent the array capacitor electrodes; and *C*_19_ represents the ring capacitor electrode. The droplet movement will cause the capacitance values of $C_{1}^{19}$$C_{2}^{19}$, …, $C_{18}^{19}$ in the mutual capacitance matrix of the sensor changing. The position of the droplet was determined by detecting the change in mutual capacitance, and then the output angle signal was calculated.

### 2.3. Design of Sensor Capacitor Electrodes

The measurement of the angle signal by a pair of capacitor electrodes is limited. Therefore, the number of capacitor electrodes needs to be increased to improve the measurement range of the sensor. The design of the sensor capacitor electrodes needs to meet the following principles. First, the capacitor electrodes need to be in the same plane to facilitate MEMS fabrication. Second, the rate change of the mutual capacitance when the droplet passes the capacitor is large, and finally, the number of array capacitor electrodes should be as small as possible. Combining the above principles, two pairs of capacitor electrodes were placed on both sides of the metal droplet movement groove, as shown in Figure 5. The dimensions of the capacitor electrodes 1, 2, 3, and 4 in Figure 5 were all 0.5 mm × 0.5 mm, and the diameter of the metal droplet was 1.8 mm. We used COMSOL numerical simulation software to perform the mutual capacitance simulation, calculate the mutual capacitance between capacitor electrodes 1 and 2, and the mutual capacitance between capacitor electrodes 3 and 4 when the mercury droplet moved from point A to point B. Note that Point A and Point B were 10 mm apart).

Capacitor electrodes 1, 3, and capacitor electrodes 2, 4 were 1.5 mm apart as shown in Figure 5. The mutual capacitance matrix of the capacitor system *C_system_* formed by the four capacitor electrodes is shown in Equation (6):(6)$$C_{system}=\left[\begin{array}{cccc} C_{11} & C_{12} & C_{13} & C_{14}\\ C_{21} & C_{22} & C_{23} & C_{24}\\ C_{31} & C_{32} & C_{33} & C_{34}\\ C_{41} & C_{42} & C_{43} & C_{44} \end{array}\right]$$

When the metal droplet moves from point A to point B, the droplet will affect the mutual capacitance value between the capacitor electrodes on both sides of the movement direction, and the sensor needs to measure the mutual capacitance between the capacitor electrodes on both sides. Therefore, only the mutual capacitance between capacitor electrodes 1, 3, and 2, 4 is concerned in the model, that is, the values of *C*_12_ and *C*_34_ in the mutual capacitance matrix Equation (6) (*C*_12_ = *C*_21_, *C*_34_ = *C*_43_). Furthermore, the distance between the capacitor electrodes on the same side changed, and the distance between the adjacent capacitor electrodes and the position of the metal droplets were studied. The results are shown in Figure 6.

Figure 6 is the relationship between the droplet displacement and the mutual capacitance when the distances between the adjacent capacitor electrodes were 1.5 mm, 2 mm, 2.5 mm, and 4 mm, respectively. When the droplet moves from point A to point B, it first approaches the capacitor formed by the 1, 2, capacitor electrodes, so *C*_12_ changes first and increases from 0.95 × 10^−3^ pF to 2.1 × 10^−3^ pF. As the displacement of the droplet continues to increase, the mercury droplet gradually moves away from capacitor electrodes 1, 2, and close to capacitor electrodes 3, 4. At this time, the mutual capacitance of capacitor electrodes 1 and 2 gradually decreases, while the mutual capacitance *C*_34_ gradually increases. Since the two pairs of capacitor electrodes in the model have the same size, the amount of each mutual capacitance change is the same. Further analysis found that the horizontal distances between the peaks and valleys of curves *C*_12_ and *C*_34_ were the same, both of which were 3.5 mm. This shows that the resolution distance of a pair of capacitor electrodes at this size is 3.5 mm. Therefore, increasing the number of capacitor electrodes and controlling the distance between two adjacent capacitor electrodes can effectively increase the measurement range of the sensor. To facilitate the measurement of the sensor, the distance between capacitor electrodes 1,2 and capacitor electrodes 3,4 should not exceed the period length of the curve (0~3.5 mm).

Combined with the above analysis, the capacitor electrode design of the droplet MEMS inclinometer sensor based on the principle of mutual capacitance was proposed, as shown in Figure 7. The capacitor system includes a ring capacitor electrode in the inner circle and 18 array capacitor electrodes in the outer circle. The width of the ring capacitor electrode was 0.5 mm. The size of the array capacitor electrodes was 0.5 mm × 0.5 mm, and the angle between two adjacent array capacitor electrodes was 20°. The inner circle radius of the ring capacitor electrode was 5 mm, and the inner circle radius of the array capacitor electrode was 6.5 mm.

COMSOL numerical simulation software was used to perform an analysis of the mutual capacitance changes in the designed capacitor system. A mercury droplet with a diameter of 1.8 mm moved from point A to point B, as shown in Figure 7, which represented the sensor rotated 180° clockwise in the horizontal plane. In the numerical simulation, the mercury droplet was spherical and tangent to the lower plane of the substrate, so that it was not in contact with the capacitor electrodes. Table 1 shows the properties of the material:

The initial boundary conduction of the model was as follows:The potential of the terminal array capacitor was 1 V.The potential of the terminal ring capacitor was 0 V.Metal droplet was the terminal with charge 0 C.The lower surface of the lower substrate was grounded.

The special mesh refinement process is required for the region of the mercury droplet. A mesh consisting of 36,630 elements was generated using the free tetrahedral meshing method and the average cell quality of the mesh reached 0.62.

Table 2 shows the mutual capacitance between the ring capacitor electrode (No. 11) and array capacitor electrodes (No. 1 to No. 10) in the sensor when the droplet stays at point A.

It can be seen from Table 2 that when the droplet stayed at point A, the mutual capacitance between the No. 1 array capacitor electrode and the No. 11 ring capacitor electrode was the largest, and the amount of mutual capacitance was 7.26 × 10^−3^ pF. Furthermore, the mutual capacitance between the array capacitor electrodes (No. 2 to No. 10) and the ring capacitor electrode gradually decreased and approached a fixed value of 3.5 × 10^−3^ pF. Table 3 shows the relationship between the angle and the mutual capacitance of the No. 1 ring capacitor electrodes. (the input angle was from 0° to 15°)

Figure 8 shows the amount of mutual capacitance between the array capacitor electrodes (No. 1 to No. 10) and the ring capacitor electrode (No. 11) when the position of the droplet changes from point A to point B (calculating the mutual capacitance every 5°). The mutual capacitance between the array capacitor electrodes and the ring capacitor electrode will increase when the droplet is close to the array capacitor electrode, and the amount of mutual capacitance can be increased from 3.4 × 10^−3^ pF to 7.3 × 10^−3^ pF. After that, the mutual capacitance reduces to 3.4 × 10^−3^ pF again as the mercury droplet gradually moves away from the array capacitor electrode. Additionally, with this geometry, the change rate of the mutual capacitance of the sensor can reach or even exceed 100%, which can greatly promote signal acquisition and resolution during testing.

### 2.4. Design of the Superhydrophobic Structure on Lower Structure

The minimum measurement angle of the MEMS liquid inclinometer sensor is mainly determined by the sliding angle of the metal droplet [12]. In order to further reduce the sliding angle of the droplet, researchers have proposed a method for growing microstructures on the surface of the substrate, which has increased the surface roughness of the substrate and reduced the contact area of the droplet and the surface [21,22]. Lv manufactured an array of micro-cubes to create a super hydrophobic surface when predicting the sliding angle of a droplet on a micro-structured surface. The sliding process proposed consisted of three steps using a high-speed camera to observe the movement of the droplet [23].

When the plate is inclined, the rear contact line of the droplet is detached from the microstructure first. At this time, the front contact line of the droplet does not change significantly. With the increase of the tilt angle *α*, the front contact line of the droplet contacts the micro-structure of the front end. However, the rear contact line does not change. As the inclination of the microstructure substrate further increases, the droplet will repeat the entire sliding process described above, as shown in Figure 9a. During the sliding process of the droplet, the contact line moves in only one direction at the same time. Therefore, the movement of the droplet on the surface of the micro-cube microstructure is considered to be quasi-static and the energy is conserved, as shown in Figure 9b. When the tilt angle of plate is *α*, the area where the rear contact line moves is Δ*S* and the amount of change in droplet surface energy Δ*E* at this time is shown in Equation (7):(7)$$\Delta E=(1+\cos\theta )\gamma_{lv}\Delta S$$
where *γ_lv_* is the surface tension of droplet; *θ* is the contact angle of surface and droplet; and Δ*S* is the area of liquid surface movement when rear contact line moves. The value of the Δ*S* is:(8)$$\Delta S=\Delta xR\frac{l}{\left(l+d\right)}$$
where Δ*x* is the displacement of the rear contact line movement; *R* is the radius of the contact surface of the droplet with the surface; *l* is the side length of the micro-cubes; and *d* is the distance between two adjacent micro-cubes.

When the inclination angle is *a*, the droplet slides from the super hydrophobic surface with the micro-cube structure, and the displacement of the rear contact line of the droplet is Δ*x*. At this time, the gravity center of the droplet is moved by Δ*x*/2, and the gravitational potential energy is:(9)$$\Delta E^{\prime }=mg(\Delta x/2)\sin\alpha$$

According to the law of the conservation of energy, the relationship between the sliding angle and the characteristic size of the micro-structure is as follows:(10)$$\sin\alpha =2(1+\cos\theta )\gamma_{lv}lR/mg(l+d)$$

Based on the above prediction model for the sliding angle, the micro-cube structure on the lower surface was designed (that is, the surface between array capacitor electrodes and ring capacitor electrode on the lower substrate) to reduce the sliding angle of the mercury droplet. As shown in Figure 10, the size of the micro cubes was 30 × 30 um, and the distance between each adjacent cube was 20 um.

In order to measure the sliding angle of a mercury droplet on the micro-cube structure, a layer of silica array micro-cubes with a size of 30 um × 30 um and a pitch of 20 um was deposited on the glass through the MEMS process. Next, a 1.8 mm diameter mercury droplet was dropped on the glass and the whole substrate was placed on the tilted platform, as shown in Figure 11. The tilted platform could rotate around the horizontal axis. The high-speed camera observed that when the tilt angle was 2.12°, the droplet started to slide, which shows that the minimum measurement angle of the sensor is 2.12° and the measurement range is from ±2.12° to ±360°.

## 3. Fabrication Process of a Typical Sensor

Based on the theoretical analysis of the sensor design, the fabrication process of the MEMS liquid droplet inclinometer sensor was proposed. The process mainly includes the manufacture of the lower substrate and the intermediate substrate as well as the packaging process after injecting the mercury droplet, as shown in Figure 12a. The metal capacitor electrodes, silicon dioxide insulating layers, and super-hydrophobic microstructures were fabricated on the surface of the lower substrate. The super hydrophobic micro-structures were silica cubes, and the side length of the cube was 30 um. The distance between each adjacent cube was 20 um, and the thickness of the cube was the thickness of the silicon dioxide layer, which was 300 nm. For the intermediate substrate, the width of the annular groove was 2 mm, and the depth was 2 mm. Furthermore, the diameter of the mercury droplet was 1.8 mm. The detail processed is detailed as follows.

First, a layer of photoresist (EPG 535) was sprayed on the surface of a 4-inch size Plexiglas substrate (the material of the lower substrate was Plexiglas to avoid the influence of the semiconductor properties of the silicon wafer on the capacitance of the sensor). After that, the photomask was used in this step to expose the array capacitor electrodes, and the ring capacitor electrode pattern on the lower substrate. A Cr/Au metal film with a thickness of about 150 nm was sputtered onto the wafer substrate by the radio frequency (RF) method using a magnetron sputtering machine to form the metal capacitor films. Second, a silicon dioxide layer with a thickness of 300 nm was deposited on the surface of the prepared metal capacitor electrodes by a plasma-enhanced chemical vapor deposition (PECVD) process. A layer of photoresist film (EPG 535) was coated on the lower substrate, and a second photomask was used in this step to form the micro-cube pattern to protect the metal capacitor electrodes. Finally, a wet micro-etching process was used to form the hydrophobic micro-cube structure through the silicon dioxide layer.

SU-8 photoresist was used to make a 2 mm deep annular groove on the intermediate substrate. First, 23 g photoresist was slowly poured on the lower structure, then the whole wafer was placed on a horizontal plate and left for 10 h. This step replaces the coating step so that the photoresist can spread evenly on the substrate surface. The photoresist is soft baked through step-by-step heating, with a heating speed of 1 min per 1 °C where 65 °C is the first step for a duration of 30 min and 95 °C is the second step for 60 h (step-by-step cooling is ensured after the end of the soft bake). Next, the substrate was exposed by a photolithography machine with an exposure dose of 2240 mj/cm^2^. The exposed photoresist was post-baked to complete the exposed process of photoresist. The post-baking temperature was 95 °C and the time was 1 h. The heating method used was also step-by-step heating. Finally, the photoresist was developed to obtain a 2 mm annular groove, and the development time was four hours.

After completing the process of the intermediate substrate, first, the chips needed to be diced. Then, a mercury droplet with a diameter of 1.8 mm was injected into the annular groove. Finally, the upper layer and the intermediate substrate were attached by UV adhesive. The material of the upper layer was glass, and the packaging process was under normal temperature and pressure. The size of the upper substrate was the same as that of the intermediate substrate. The droplet needs to ensure that the pressure in the left and right ends of the annular groove is the same during the movement. Hence, the incomplete sealing is conducive to the sliding of the droplet. Figure 12b shows the schematic diagram of the final processed sensor. The overall size of the sensor chip was 21 mm × 21 mm.

## 4. Results and Discussion

### 4.1. Static Capacitance Test of the Sensor

The simulation design of the sensor shows that the variation range of mutual capacitance was 3.9 × 10^−3^ pF. Therefore, a proper signal acquisition board is needed to test the capacitance signal. The AD7747 commercial capacitance evaluation board from Analog Devices (ADI) was used to test the initial mutual capacitance of the sensor [24]. The evaluation board has two input ports, CIN1+/CIN1− and GND. Furthermore, a USB cable was used to connect the PC, which provides the power supply. As shown in Figure 13, the array capacitor lead of the sensor was connected to CIN1+, and the ring capacitor lead was connected to GND. The signal acquisition frequency was selected to be 45.5 Hz. The initial mutual capacitances between five array capacitor electrodes and the ring capacitor electrode were measured respectively. The mercury droplets were far away from each array capacitor electrode. The results are shown in Table 4.

Table 4 shows the initial mutual capacitance values between the five array capacitor electrodes and the ring capacitor electrode. The mutual capacitance values were approximately the same and larger than the designed initial mutual capacitance values (3.5 × 10^−3^ pF). In the design of the sensor, the distance between the array capacitor and ring capacitor as well as the materials and sizes of the capacitor electrodes were the same, so the initial mutual capacitance should be similar. Due to the parasitic capacitance between the test circuit board, the connection port, and the wires, the mutual capacitance value obtained from the test was greater than the designed mutual capacitance amount. Hence, in the dynamic test of the sensor, it was necessary to fix the wires and the circuit board to ensure that the parasitic capacitance will not change and affect the output signal of the mutual capacitance.

### 4.2. Dynamic Signal Test of the Sensor

To test the dynamic performance of the sensor, the experimental system was placed on a tilted test bench, as shown in Figure 14. The experimental test bench consists of an indexing head capable of biaxial rotation, an angle display, a capacitance test circuit, and a PC host computer. The sensor was fixed on the test platform of the indexing head. Initially, the angle between the sensor and the horizontal plane was 75° and the mercury droplet stayed at the lowest point in the annular groove of the intermediate substrate due to gravity (the array capacitor electrode here was No. 1). The indexing head was turned to make positive and negative rotations around the axial direction. Meanwhile, the evaluation board and connection wires were fixed during the test process to eliminate the influence of parasitic capacitance on mutual capacitance. The signal acquisition frequency of the AD7747 was selected as 45.5 Hz, and the acquisition time was 1 min. Table 5 shows the relationship between the angle and the mutual capacitance obtained by the host computer processing software of the AD7747 (as the angle between the initial position of the sensor and the horizontal plane was 75°, the output angle needed to be calibrated using Equation (4)).

It can be seen from Table 5 that when the input angle was 0°, the droplet was located in the middle of the array capacitor electrode No. 1 and the ring capacitor electrode, so the mutual capacitance value of this capacitor reached the maximum amount of 0.705 pF. As the indexing head rotated, the droplet position changed and gradually deviated from array capacitor electrode No. 1. When the input angle reached 10°, the mutual capacitance value reached the same as the initial capacitance in Table 4 (the value here was around 0.680 pF), and as the input angle continued to increase, the mutual capacitance value between array capacitor electrode No. 1 and the ring capacitor electrode no longer changed.

Figure 15 shows the mutual capacitance value between array capacitor electrodes Nos. 1, 2, 3, and the ring capacitor electrode when the droplet moved from 0° to 45° (capacitor electrode No. 1 corresponded to 0°; capacitor electrode No. 2 corresponded to 20°, and capacitor electrode No. 3 corresponded to 40°).

It can be seen from Figure 15 that when the droplet approached an array capacitor electrode, the mutual capacitance values between this capacitor electrode and the ring capacitor electrode started increasing. This trend is consistent with the design of the sensors. The mutual capacitance value was around 0.706 pF when the droplet was in the middle of the capacitor electrode, and the mutual capacitance value was 0.679 pF when the droplet was away from the capacitor electrode. Comparing the change in mutual capacitance between the capacitor electrodes in the numerical simulation results and test results, the mutual capacitance measured by the dynamic test was larger (the amount of change was 2.5 × 10^−2^ pF). This is because the numerical analysis model only included the capacitor electrodes and did not consider the metal pads. Furthermore, in the numerical simulation, due to the limitation of the meshing process in the COMSOL simulation software, the capacitor electrodes were regarded as a two-dimensional surface, and the thickness was ignored (the real capacitance value was greater than the numerical simulation results as shown in Table 4). The value of the measured capacitance when the mercury droplet was in the middle of the two capacitor electrodes is the series capacitance system of the capacitance formed by the mercury droplet and the two electrodes. Therefore, the real capacitance change value was greater than the results of the numerical simulation. Meanwhile, mercury droplets are not standard spherical when moving or stationary in the annular groove. The minimum resolution of the AD7747 capacitance evaluation board was 1 × 10^−3^ pF, and when the input angle changed from 0° to 10°, the changing value of the mutual capacitance was 2.5 × 10^−2^ pF. Therefore, the resolution of the sensor was 0.4°.

## 5. Conclusions

This paper proposed a liquid droplet MEMS inclinometer sensor and describes the structural design and manufacturing process of the sensor in detail. The sensor consisted of an upper layer, a lower substrate, an intermediate substrate, and a mercury droplet as the sensitive element. A ring capacitor electrode and 18 array capacitor electrodes were deposited on the lower substrate through a MEMS processing process. The intermediate substrate consisted of an annular groove that provided a moving channel for the mercury droplet. The sensor expressed the output angle by the change in the mutual capacitance between the ring capacitor electrode and the array capacitor electrodes.

Static and dynamic tests of the sensor signals were performed. The test results showed that the capacitance between the 18 array capacitor electrodes and the ring capacitor electrode of the sensor was similar and relatively stable in the static test. Furthermore, in the dynamic test of the sensor, the value of the mutual capacitance increased when the liquid droplet approached the capacitor electrode and decreased when it was far away from the capacitor electrode, which is the same as that designed. The measurement range of the sensor was from ±2.12° to ±360°, and the resolution of the sensor was 0.4°. The research further expands the measurement range of the liquid droplet MEMS inclinometer sensors, and innovatively solves the issue where the previous droplet sensor can only measure special angles.

## Figures and Tables

**Figure 1 sensors-20-03711-f001:**
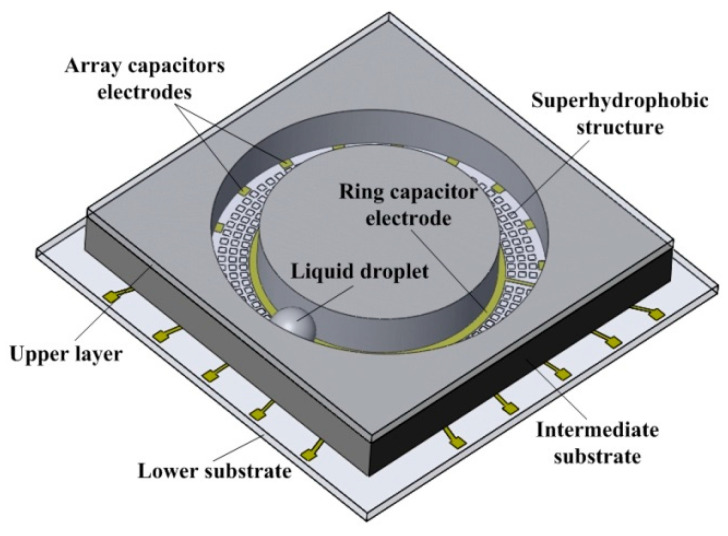
A schematic diagram of the microelectromechanical system (MEMS) liquid droplet inclinometer sensor structure.

**Figure 2 sensors-20-03711-f002:**
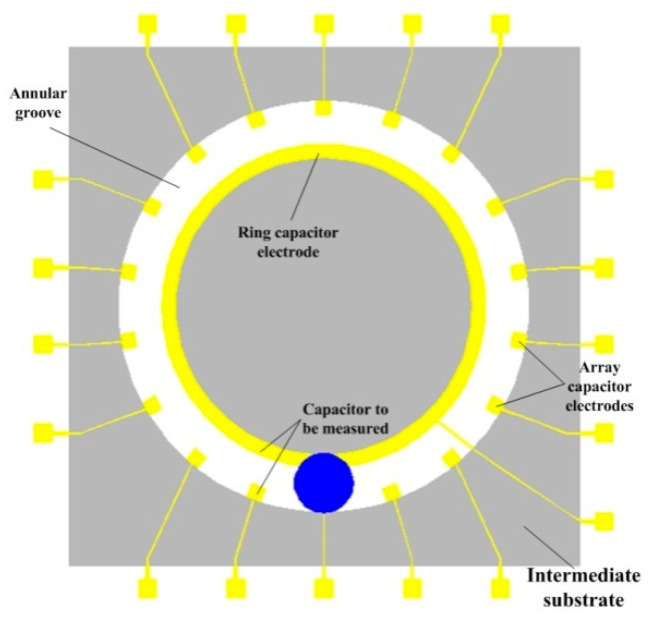
The schematic diagram of the array capacitor plates and the ring capacitor plate.

**Figure 3 sensors-20-03711-f003:**
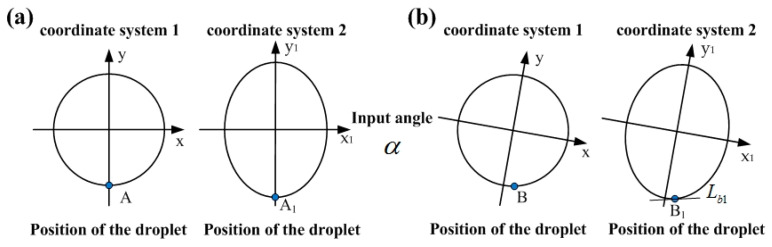
The position of the droplet in the annular groove. (**a**) Initial position, (**b**) position after input angle.

**Figure 4 sensors-20-03711-f004:**
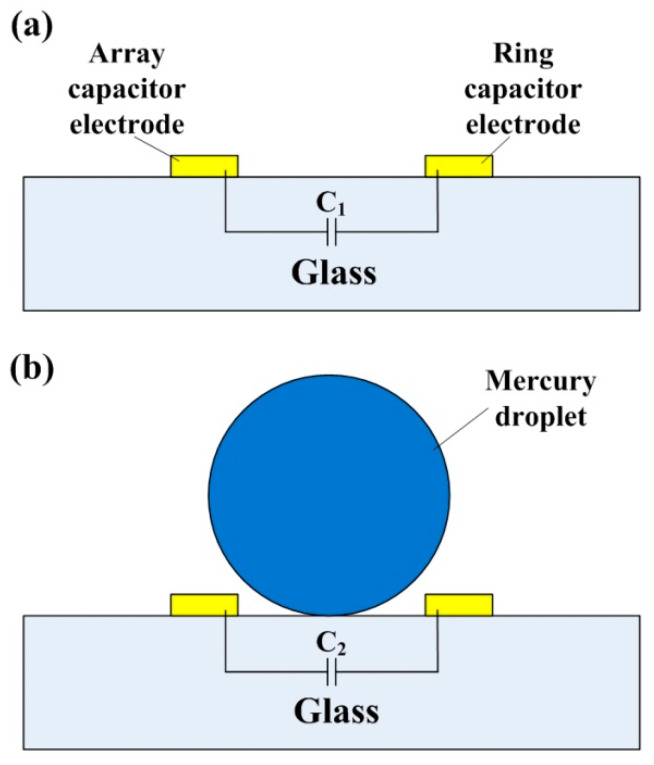
The schematic diagram of the mercury droplet and capacitors. (**a**) Droplet far away from capacitors, (**b**) droplet close to the capacitors.

**Figure 5 sensors-20-03711-f005:**
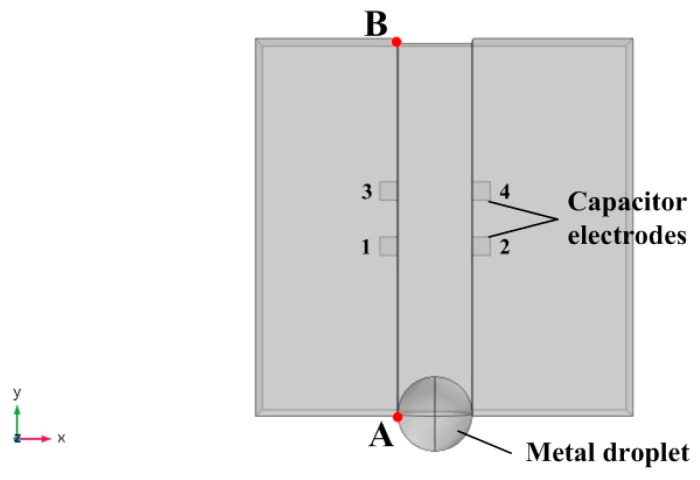
Schematic diagram of two pairs of capacitor electrodes.

**Figure 6 sensors-20-03711-f006:**
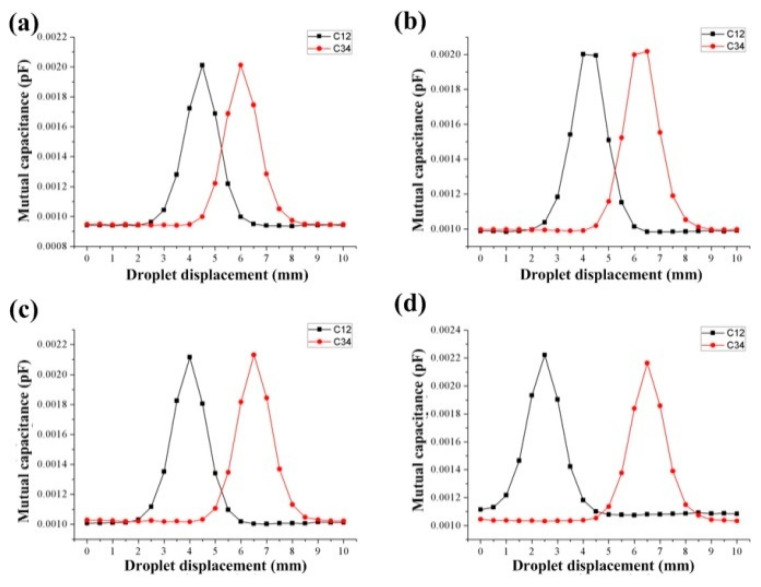
(**a**–**d**) The relationship between the distance of the adjacent capacitance electrodes and the position of the droplet.

**Figure 7 sensors-20-03711-f007:**
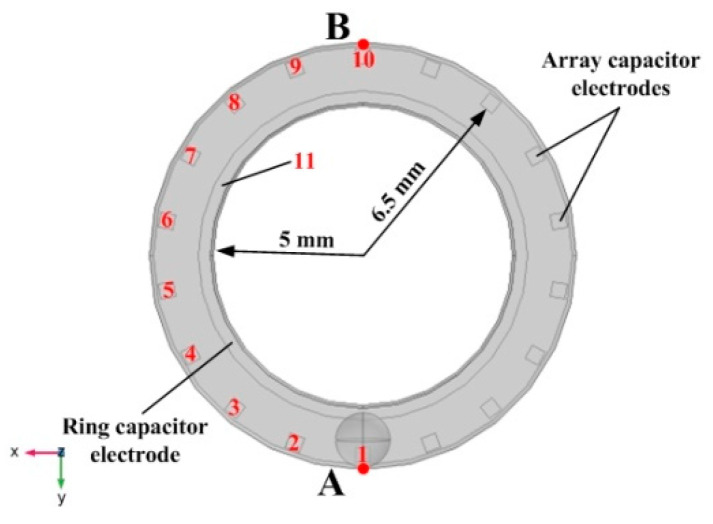
The capacitor electrodes of the sensor.

**Figure 8 sensors-20-03711-f008:**
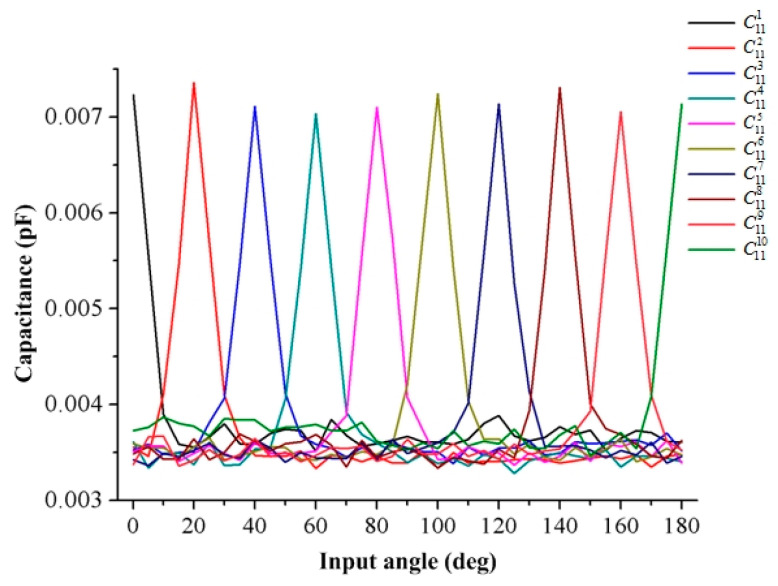
The amount of the mutual capacitance when the mercury droplet moves from Point A to Point B.

**Figure 9 sensors-20-03711-f009:**
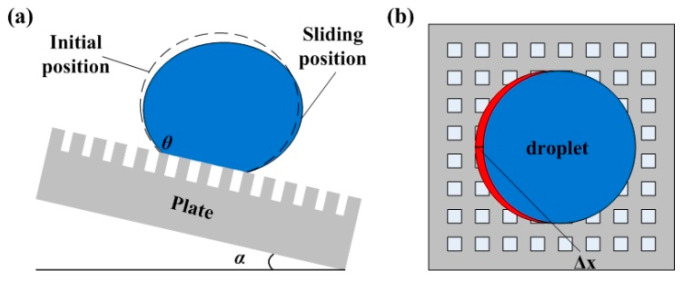
Schematic diagram of droplet sliding on the super hydrophobic surface. (**a**) is the schematic diagram of droplet sliding. (**b**) is the top view of the droplet movement.

**Figure 10 sensors-20-03711-f010:**
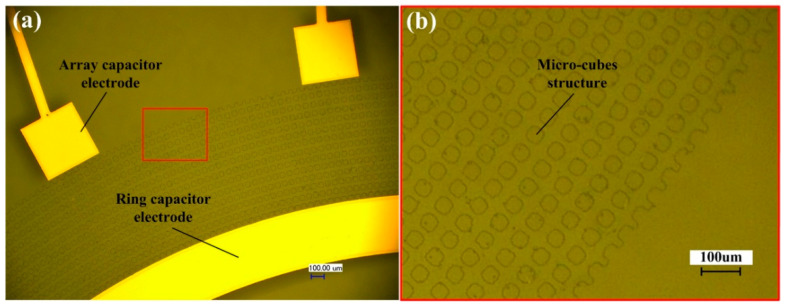
(**a**) Schematic diagram of micro-cube structure in sensors. (**b**) Detail of the micro-cube structure in the red square.

**Figure 11 sensors-20-03711-f011:**
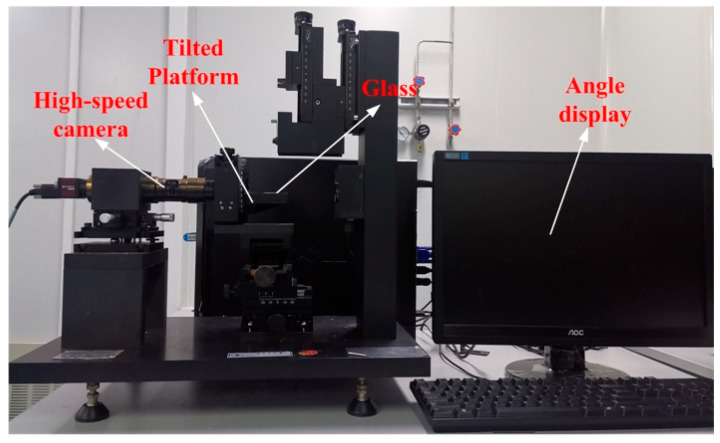
The platform of the sliding angle test.

**Figure 12 sensors-20-03711-f012:**
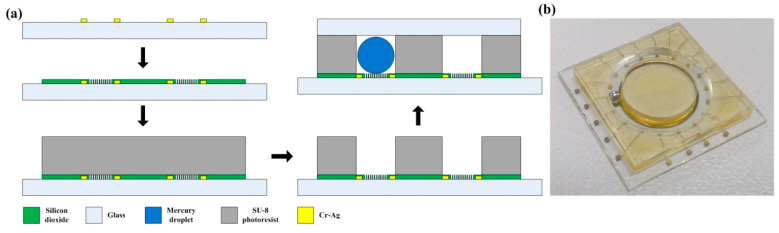
(**a**) Schematic diagram of the manufacture process, (**b**) fine processed sensor.

**Figure 13 sensors-20-03711-f013:**
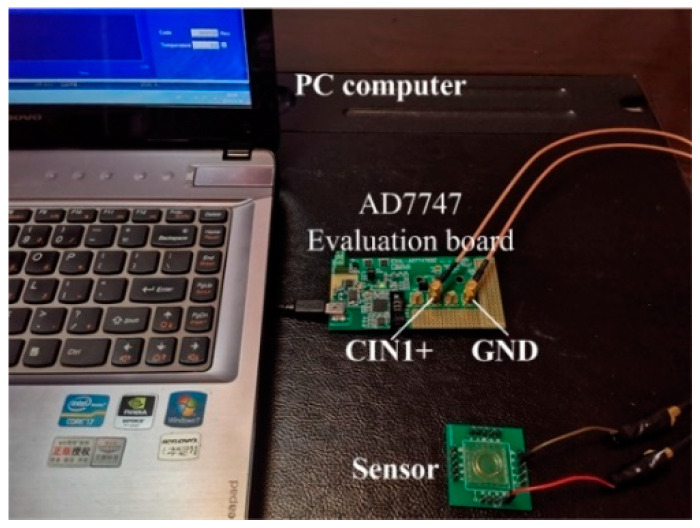
The static experiment test bench.

**Figure 14 sensors-20-03711-f014:**
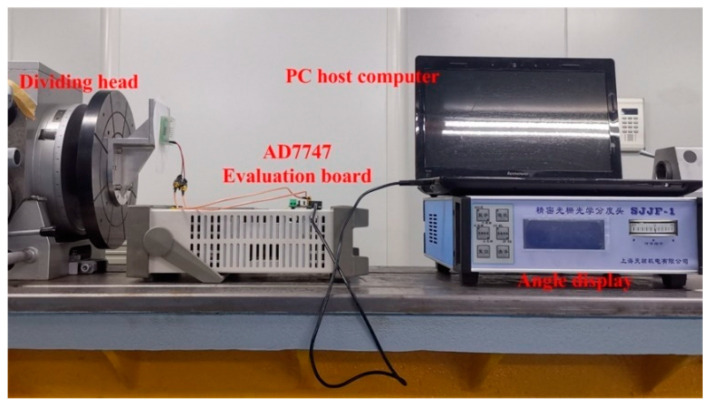
The dynamic experiment test bench.

**Figure 15 sensors-20-03711-f015:**
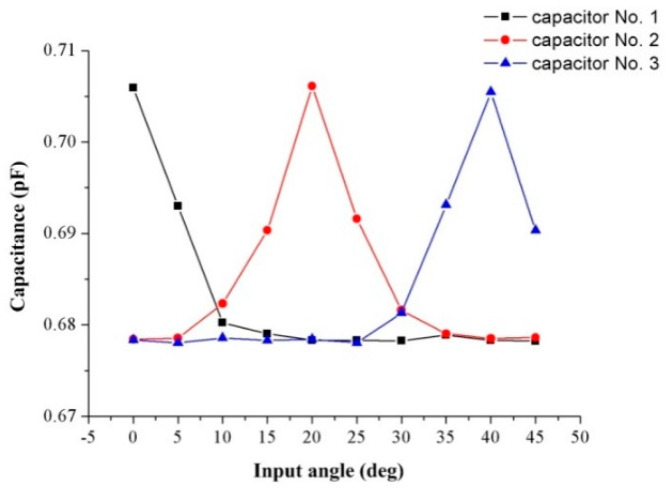
The mutual capacitance of three capacitors.

**Table 1 sensors-20-03711-t001:** The material properties of the model.

	Material	Relative Permittivity
Upper layer	glass	4.2
Lower substrate		
Liquid droplet	mercury	1
Intermediate substrate	SU-8 photoresist	4

**Table 2 sensors-20-03711-t002:** The mutual capacitance when droplet stays at point A.

Number	1	2	3	4	5	6	7	8	9	10
Capacitance (pF)	7.26 × 10^−3^	3.60 × 10^−3^	3.50 × 10^−3^	3.33 × 10^−3^	3.51 × 10^−3^	3.51 × 10^−3^	3.49 × 10^−3^	3.53 × 10^−3^	3.58 × 10^−3^	3.79 × 10^−3^

**Table 3 sensors-20-03711-t003:** The relationship between the angle and the mutual capacitance.

Input Angle (Degree)	Mutual Capacitance (pF)	Relative Capacitance (pF)
0	0.007229	0.0016 0.0016 0.0003
5	0.005564	
10	0.003902	
15	0.003587	

**Table 4 sensors-20-03711-t004:** Initial capacitance of five capacitors.

Number	1	2	3	4	5
**Initial capacitance (pF)**	0.679031	0.678423	0.679345	0.67821	0.678395

**Table 5 sensors-20-03711-t005:** The relationship between the angle and the mutual capacitance.

Input Angle (Degree)	Output Angle (Degree)	Mutual Capacitance (pF)	Relative Capacitance (pF)
0	0	0.705636	0.0086 0.0060 0.0057 0.0050 0.0003
3	2.89	0.697036	
5	4.81	0.691021	
8	7.73	0.685328	
10	9.66	0.680231	
15	14.51	0.679868	

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
