# Peer review of "A Capacitive MEMS Inclinometer Sensor with Wide Dynamic Range and Improved Sensitivity"

_sensors, 2020, doi:10.3390/s20133711_

Round 1

Reviewer 1 Report

The paper presents a capacitive MEMS inclinometer sensor with wide dynamic range and improved sensitivity. Sensor design and fabrication are given in a deep detail, also some measurements are shown. According to the results presented, the measuring range and the resolution of MEMS liquid inclinometer is highly improved.  The conversion degrees-capacitance is related to the connection mode of three capacitors (C2 and C3 in series and then in parallel with C1 ). The total capacitance of the system (not the mutual one) should be the parameter to be expressed as a function of degrees (sensitivity), I am NOT convinced about the output parameter. No comparison with literature. 

Author Response

Reviewer1:

The paper presents a capacitive MEMS inclinometer sensor with wide dynamic range and improved sensitivity. Sensor design and fabrication are given in a deep detail, also some measurements are shown. According to the results presented, the measuring range and the resolution of MEMS liquid inclinometer is highly improved.  The conversion degrees-capacitance is related to the connection mode of three capacitors (C2 and C3 in series and then in parallel with C1 ). The total capacitance of the system (not the mutual one) should be the parameter to be expressed as a function of degrees (sensitivity), I am NOT convinced about the output parameter. No comparison with literature.

We deleted the description of Ctotal, the connection of C1, C3, and C2, as well as Equation 5 in section 2.2. Besides, we also modified Figure 4 to show the process of droplets approaching the capacitor electrodes more clearly. The revised picture is shown below:

Figure 4. The schematic diagram of mercury droplet and capacitors. (a) is the droplet far away from capacitors, (b) is the droplet close to the capacitors.

The droplet passing between the two capacitor electrodes is equivalent to reducing the distance between the ring capacitor and the array capacitor. So that the mutual capacitance of the ring capacitor and the array capacitors increase as the mercury droplet approaches to the capacitor electrodes and decreases as the mercury droplet moves away from the capacitor electrodes.

Hence, as the metal droplets gradually approach the capacitor electrodes, the mutual capacitance between the two plates gradually increases from C1 to C2 as shown in Figure 4. The test capacitance value of the sensors is the mutual capacitance of the ring capacitor and the array capacitors. Therefore, this principle can be used to design the sensor. (Due to the complexity of the capacitor system, the finite element simulation method was used to calculate the change in mutual capacitance of sensor. The results are shown in Figure 8 in the manuscript).

Reviewer 2 Report

The manuscript is well written. It presents a novel MEMS inclinometer sensor, from design, simulation, fabrication and testing perspectives. The organization of the manuscript is clear, and contents are intriguing. 

The following questions need to be addressed before acceptance:

  1. Section 2.3 first analyzed the mutual capacitance relation between two pairs of electrodes. However, simulation is also carried out on the real model, instead of the simplified one. Isn't it more reasonable to show results directly on the real model? Does the 3.5 um spacial resolution still hold true for the real model?
  2. It may not be accurate to use the term 'dynamic analysis of mutual capacitance changes' to describe the capacitance change analysis with the location change of the conductive droplet, since it is a static analysis in nature. 
  3. The 'micro cubes' mentioned multiple times in the manuscript isn't very clear to the readers. Schematic and labels of dimensions are needed for illustration.
  4. Elaboration is needed for how the 2.12 degree sliding angle is obtained for a mercury droplet with a diameter of 1.8 mm.
  5. It may worth elaborating how the upper layer is covered to complete the fabrication. Is it completely sealed and how?
  6. Table 3 only shows the results when the input angle is up to 15 degree. Similarly, the maximum input angle in Figure 14 is 45 degree. It is not convincing enough to claim the measurement range of the sensor is up to 360 degree. Could the authors show more test data to prove the claim.
  7. How is the repeatability of the measurement? Is calibration required for each measurement?
  8. Since parasitic capacitance is mentioned as the main source that causes the big discrepancy between the measurement and simulation, it will worth showing capacitance change (delta C) instead of the absolute value. Please also correlate the simulation results and test results and explain the discrepancy. 

Author Response

Reviewer2:

The manuscript is well written. It presents a novel MEMS inclinometer sensor, from design, simulation, fabrication and testing perspectives. The organization of the manuscript is clear, and contents are intriguing. The following questions need to be addressed before acceptance:

  1. Section 2.3 first analyzed the mutual capacitance relation between two pairs of electrodes. However, simulation is also carried out on the real model, instead of the simplified one. Isn't it more reasonable to show results directly on the real model? Does the 3.5 um spacial resolution still hold true for the real model?

The mutual capacitance between the two pairs of electrodes is mainly affected by the size of the capacitance electrodes and the distance between each electrode. The simplified model is to study the relationship between the mutual capacitance of the array capacitor with a size of 0.5mm×0.5mm and the position of the droplet. The results show that the spatial resolution at this size is 0~3.5mm, so the design of the number of real sensor array capacitor electrodes needs to meet the above conditions. In our design, the angle between two adjacent array capacitor electrodes is 20 ° and the center distance between two adjacent capacitor electrodes is 2.35mm (the radius is 6.75mm). If the designed angle is 30° (the number of electrodes is 12), the center distance between two adjacent capacitor electrodes is 3.53mm. Besides, if the angle between two adjacent capacitor electrodes is 15°, the center distance between the two adjacent capacitor electrodes can meet the requirements, but the number of electrodes needs to be increased to 24. We use the real model to simulate the different distances between adjacent two capacitor electrodes (The angle is 30°). The results are as follows:

As can be seen in the figure, the real model results meet the trend of spatial resolution (0~3.5mm). Therefore, in order to facilitate simulation and display results, a simplified model is used to design the number of array capacitor electrodes.

  1. It may not be accurate to use the term 'dynamic analysis of mutual capacitance changes' to describe the capacitance change analysis with the location change of the conductive droplet, since it is a static analysis in nature.

We deleted the word ‘dynamic’, and the modified sentence now is ‘COMSOL numerical simulation software is used to perform analysis of mutual capacitance changes in the designed capacitor system’.

  1. The 'micro cubes' mentioned multiple times in the manuscript isn't very clear to the readers. Schematic and labels of dimensions are needed for illustration.

We modified Figure 10 and involved the detailed information of the Micro-cubes structure in the picture. The figure is shown as follow:

Figure 10. (a) is the schematic diagram of micro-cubes structure in sensors. (b) is the detail of the micro-cubes structure in the red square.

  1. Elaboration is needed for how the 2.12 degree sliding angle is obtained for a mercury droplet with a diameter of 1.8 mm.

We added the description of experimental test: “In order to measure the sliding angle of a mercury droplet on micro-cubes structure, a layer of silica array micro-cubes with a size of 30×30um and a pitch of 20um is deposited on the glass through the MEMS process. After that, a 1.8mm diameter mercury droplet is dropped on the glass and the whole substrate is placed on the tilted platform as shown in Figure 11. The tilted platform can rotate around the horizontal axis. The high-speed camera observes that when the tilt angle is 2.12°, the droplet starts to slide, which shows that the minimum measurement angle of the sensor is 2.12° and the measurement range is from ±2.12° to ±360°.”

Figure 11. The platform of sliding angle test.

  1. It may worth elaborating how the upper layer is covered to complete the fabrication. Is it completely sealed and how?

We modified the packaging process, and the packaging process of the upper substrate is described:

After completing the process of the intermediate substrate, the chips need to be diced first. Then, a mercury droplet with a diameter of 1.8 mm is injected into the annular groove. Finally, the upper layer and the intermediate substrate are attached by UV adhesive (The material of the upper layer is glass, and the packaging process is under normal temperature and pressure). The size of the upper substrate is the same as that of the intermediate substrate. The droplets need to ensure that the pressure in the left and right ends of the annular groove is the same during the movement. Hence, the incomplete sealing is conducive to the sliding of the droplet.

  1. Table 3 only shows the results when the input angle is up to 15 degree. Similarly, the maximum input angle in Figure 14 is 45 degree. It is not convincing enough to claim the measurement range of the sensor is up to 360 degree. Could the authors show more test data to prove the claim.

Thank you for your comments. In fact, during the research process, the sensor is fixed on the platform of the indexing head and the angle between the rotating platform and the horizontal plane is 75°. Therefore, the actual maximum test range of the sensor is calculated as ±347.7° by Equation 4. The maximum test range of 360° is an ideal state. That is when the angle between the sensor and the horizontal plane is 90°. In this case, the droplet will leave from the lower substrate and cause an error in the sensor signal. The sensor needs to ensure that the angle with the horizontal is less than 90° during the measurement process.

It can be seen from Table 3 that the capacitor electrodes manufactured by the MEMS process tend to be the same in value. Therefore, the changing capacitance of the droplet as it passes through each capacitor electrode should also be the same. The experiment of 45° can reflect the sensor test range results. At present, we use the AD7747 capacitance test board for testing in the change capacitance. Only one pair of capacitor electrodes can be tested at a time. Multiple disassembly and assembly will bring the influence of parasitic capacitance. In the subsequent research, we need to develop a switching circuit to achieve multi-channel capacitance test.

  1. How is the repeatability of the measurement? Is calibration required for each measurement?

In the dynamic test of the sensor, first fix the sensor on the platform of the indexing head, and rotate the indexing head to make the angle between the rotating platform and the horizontal plane be 75°. At this time, the sensor will stay at the lowest point in the annular groove due to gravity. Since the movement trajectory of the droplet is a closed ring, the sensor needs to be zero-calibrated before the measurement. (For example, in the dynamic test experiment of the manuscript, the No. 1 capacitor electrode at the lowest point is set to 0°. After each measurement process, the sensor needs to be restored to the zero position and then tested).

  1. Since parasitic capacitance is mentioned as the main source that causes the big discrepancy between the measurement and simulation, it will worth showing capacitance change (delta C) instead of the absolute value. Please also correlate the simulation results and test results and explain the discrepancy

We modified the Table 3 and added the Table of the relative capacitance (delta C) data to show the simulation and experimental results.

The capacitance change of experimental results is large than simulation results. This is because the numerical analysis model only includes the capacitor electrodes and does not consider the metal pads. Besides, in the numerical simulation, due to the limitation of the meshing process in the COMSOL simulation software, the capacitor electrodes are regarded as a two-dimensional surface, and the thickness is ignored (the real capacitance value is greater than the numerical simulation results as shown in Table 4). The value of the measured capacitance when the mercury droplet is in the middle of the two capacitor electrodes is the series capacitance system of the capacitance formed by the mercury droplet and the two electrodes. Therefore, the real capacitance change value is greater than the results of numerical simulation. Meanwhile, mercury droplets are not standard spherical when moving or stationary in the annular groove.

Reviewer 3 Report

The Paper presents a capacitive MEMS inclinometer sensor with wide dinamic range and improved sensitivity.

It represents an extension of the previous work published by the Authors [12], but it presents a sufficient level of novelty to be published after a minor revision.

1- The English must be strongly revised. Moreover, measurements units are not always present, Table 1 is cited as table 2. The Authors should carefully check the paper.

2- The Authors should explain better how they performed numerical analyses to allow the interested reader to reproduce their results. Boundary conditions, mesh, coupled physics, hypotheses must be clearly stated in the numerical section

Author Response

Reviewer3:

The Paper presents a capacitive MEMS inclinometer sensor with wide dinamic range and improved sensitivity.It represents an extension of the previous work published by the Authors [12], but it presents a sufficient level of novelty to be published after a minor revision.

  1. The English must be strongly revised. Moreover, measurements units are not always present, Table 1 is cited as table 2. The Authors should carefully check the paper.

We have carefully revised the typos in the manuscript:

  1. “It can be seen from Table 5 that when the input angle is 0°,”
  2. “Here, the shape of the annular groove was designed as a standard circle with a radius of 1 to facilitate calculation.”
  3. “Besides, the distance between the capacitor electrodes on the same side was changed,”

….

Besides, we have modified the reference to the Equations, Tables and Figure.

  1. The Authors should explain better how they performed numerical analyses to allow the interested reader to reproduce their results. Boundary conditions, mesh, coupled physics, hypotheses must be clearly stated in the numerical section

We added the material properties, boundary conditions and mesh process of the model.

Table 1 shows the properties of the material:

Table 1. The material properties of the model

Material

Relative permittivity

Upper layer

glass

4.2

Lower substrate

Liquid droplet

mercury

1

intermediate

SU-8 photoresist

4

The initial boundary conduction of the model is:

  1. The potential of the terminal array capacitor is 1 V.
  2. The potential of the terminal ring capacitor is 0 V.
  3. Metal droplet is the terminal with charge 0 C.
  4. The lower surface of the lower substrate is grounded.

The special mesh refinement process is required for the region of the mercury droplet. A mesh consists of 36630 elements was generated using the free tetrahedral meshing method and the average cell quality of the mesh reaches 0.62.
